# The role of vicariance and dispersal on the temporal range dynamics of forest vipers in the Neotropical region

Matheus Pontes-Nogueira[1¤]*, Marcio Martins[2], Laura R. V. Alencar[3], Ricardo J. Sawaya[4]

**1** Graduação em Ciências Biológicas, Universidade Federal de São Paulo, Diadema, Brazil, **2** Departamento de Ecologia, Instituto de Biociências, Universidade de São Paulo, São Paulo, Brazil, **3** Department of Biological Sciences, Clemson University, Clemson, South Carolina, United States of America, **4** Centro de Ciências Naturais e Humanas, Universidade Federal do ABC, São Bernardo do Campo, Brazil

¤ Current address: Programa de Pós-Graduação em Evolução e Diversidade, Universidade Federal do ABC, São Bernardo do Campo, Brazil

* mpnogueira26@gmail.com

**Data Availability Statement:** All relevant data are within the manuscript and its Supporting Information files.

## Abstract

The emergence of the diagonal of open/dry vegetations, including Chaco, Cerrado and Caatinga, is suggested to have acted as a dispersal barrier for terrestrial organisms by fragmenting a single large forest that existed in South America into the present Atlantic and Amazon forests. Here we tested the hypothesis that the expansion of the South American diagonal of open/dry landscapes acted as a vicariant process for forest lanceheads of the genus *Bothrops*, by analyzing the temporal range dynamics of those snakes. We estimated ancestral geographic ranges of the focal lancehead clade and its sister clade using a Bayesian dated phylogeny and the BioGeoBEARS package. We compared nine Maximum Likelihood models to infer ancestral range probabilities and their related biogeographic processes. The best fitting models (DECTS and DIVALIKETS) recovered the ancestor of our focal clade in the Amazon biogeographic region of northwestern South America. Vicariant processes in two different subclades resulted in disjunct geographic distributions in the Amazon and the Atlantic Forest. Dispersal processes must have occurred mostly within the Amazon and the Atlantic Forest and not between them. Our results suggest the fragmentation of a single ancient large forest into the Atlantic and Amazon forests acting as a driver of vicariant processes for the snake lineage studied, highlighting the importance of the diagonal of open/dry landscapes in shaping distribution patterns of terrestrial biota in South America.

## Introduction

The Neotropical region is of great interest for the study of biogeographic processes. It has been shown to be the most biodiverse region in the world with high levels of endemism for different groups of organisms [1], including frogs [2], reptiles [3], and birds [4]. Several hypotheses have been proposed to explain this high diversity and endemism, including the Great

**Funding:** MPN was granted by Fundação de Amparo à Pesquisa do Estado de São Paulo (FAPESP; https://fapesp.br/) with the grant numbers Proc. 2014/23677-9 and Proc. 2017/11796-1. RJS was granted by Fundação de Amparo à Pesquisa do Estado de São Paulo (FAPESP; https://fapesp.br/) with the grant number proc. 2014/23677-9 and by Conselho Nacional de Desenvolvimento Científico e Tecnológico (CNPq; https://www.gov.br/cnpq/pt-br) with the grant number 312795/2018-1. MM was granted by Fundação de Amparo à Pesquisa do Estado de São Paulo (FAPESP; https://fapesp.br/) with the grant number proc. 2018/14091-1 and by Conselho Nacional de Desenvolvimento Científico e Tecnológico (CNPq; https://www.gov.br/cnpq/pt-br) with the grant number 306961/2015-6. The funders had no role in study design, data collection and analysis, decision to publish, or preparation of the manuscript.

**Competing interests:** The authors have declared that no competing interests exist.

American Biotic Interchange [5, 6], the isolation of South America as an island during the Paleogene and the Neogene (from 60 to 10 million years ago–mya; [7, 8]) and climatic fluctuations of the Pleistocene, from 2.6 mya to 12,000 years ago [9–12]. Such climatic fluctuations have been associated with the contraction and expansion of landscapes [11, 12].

The expansion of the South American diagonal of open/dry landscapes (DODL) including the Cerrado, Caatinga and Chaco ecoregions [13] (Fig 1), has raised the question of which processes have modulated the geographical distribution patterns of species we observe today in the Neotropical region. A recurring hypothesis in the literature is that the emergence of the DODL split a single large forest in South America prior to the Oligocene period into the present Atlantic and Amazon forests [10, 14–18] (Fig 1). Most of the formation of the DODL may have occurred due to two key factors: (i) the increasing aridity during cooler and drier periods that started in the Oligocene (between 28 and 25 mya; [13]); and (ii) the late Andean uplift throughout the late Miocene and early Pliocene (from 10 to 2 mya; Fig 1. [13, 19–21]), when the Andean Mountains reached its peak, between 6 to 4 mya [22]. The latter event has been suggested to have had a critical role on the last phases of the DODL formation, as the Andean

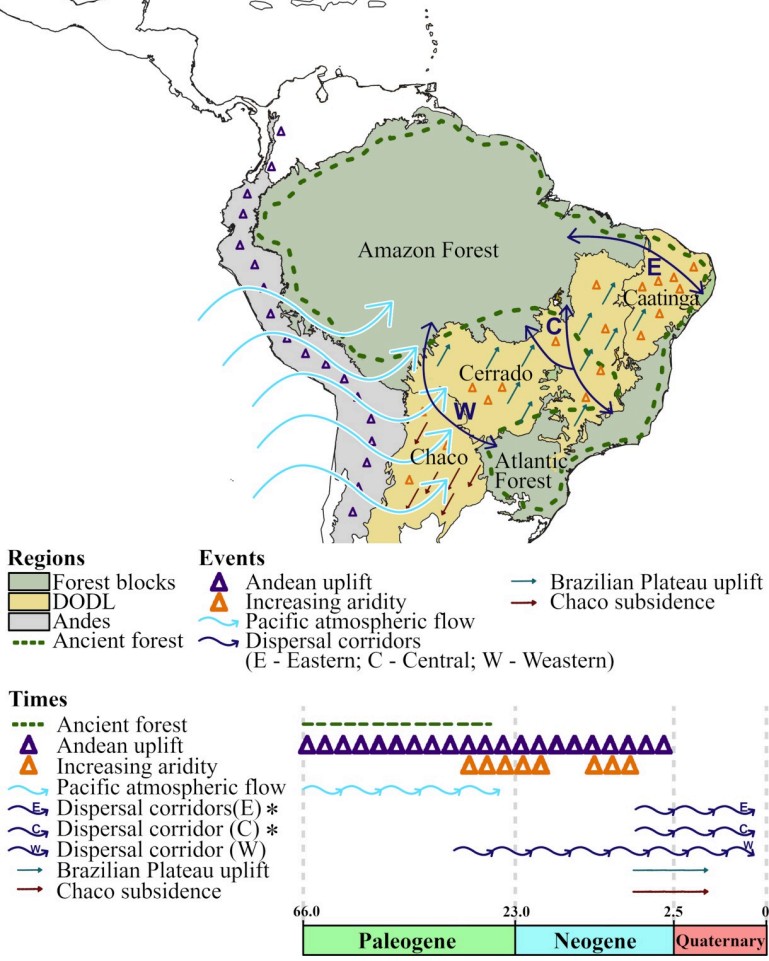

**Fig 1. Summary of important events underlying the expansion of the diagonal of open/dry landscapes (DODL).** The location of the continuous ancient forest that was subsequently splitted into the actual Amazon and Atlantic forests (dashed green line) is based on Bigarella et al. [18]. Below the map, times and periods are based on Cohen et al. [25]. The asterisks indicate times established on the literature [9, 11, 26] with recent works contesting them [27–29]. Map source: Natural Earth (www.naturalearthdata.com).

uplift led to the uplift of the Brazilian Plateau and the subsidence of the Chaco region [13, 23]. Coscaron and Morrone [24] also hypothesized that the Andean uplift may have blocked the atmospheric flow from the Pacific Ocean, helping to increase aridity and the subsequent emergence of the DODL [24].

Whether the split acted as a vicariant process for terrestrial organisms [24, 30] or created a dispersal barrier preventing populations from one forest to reach the other has been debated [16, 31]. For assassin bugs of the Peiratinae subfamily and populations of the lizard species *Polychrus marmoratus*, it was hypothesized that the emergence of the DODL acted as a vicariant process [24, 30, 32]. For the frog genus *Adenomera*, numerous dispersals followed by vicariant events between the Amazon and DODL have been suggested, and one of these events have likely occurred between the Amazon and the Atlantic Forest, forming an Atlantic forest clade [28]. The DODL has also been suggested to have isolated the frogs of the genus *Dendrophryniscus* and *Amazonella* [33]. Lizards of the genus *Enyalius* colonized the Amazon from an Atlantic Forest ancestor in the late Oligocene, which corresponded to a cladogenetic event [27].

Historical connections between the two forest blocks through dispersal corridors within the DODL have been suggested (Fig 1) [10, 11, 34]. Such corridors, presently composed of relicts of the ancient large forest block, and including gallery forests of the Cerrado and the *brejos* of the Caatinga [10, 16], changed the view that the DODL is a separated and isolated region from both the Amazon and Atlantic forests [10]. Three major corridors have been proposed (see Fig 1) [10, 11, 26, 35]. The Eastern corridor would have connected the northern Atlantic Forest and eastern Amazon through the Caatinga region; the Central corridor would have been located in central Brazil, in the Cerrado region; and the Western corridor would have been located in southwestern South America, including the Chaco, in western Brazil, Paraguay, Bolivia, and Peru [10].

Those dispersal corridors in South America were apparently present during different geological periods (Fig 1) [34]. The Eastern and Central corridors possibly have opened up more recently, in the Quaternary [14, 15, 32, 34], during forest contractions and retractions in the Plio-Pleistocene epochs (~ last 5 mya; [9, 12, 36]). A recent study, however, suggests more ancient dispersal events through these "young pathways" in the early Miocene [27], 23 to 5 mya. The Western corridor, on the other hand, is hypothesized to be older and more relevant [34, 37], despite recent discussions about its importance [35]. The emergence of this corridor was congruent to sea introgressions forming the Pebas system, between 23 and 10 mya [21, 38, 39], with the expansion of the Amazon basin between 10 and 7 mya [21, 38], and with the emergence of the Paraná sea [40, 41]. Those events may have provided paths for terrestrial organisms to disperse [42]. Other studies have elaborated further on the Western corridor, suggesting that rather than a direct link between the Atlantic and Amazon forests, the corridor would have connected the Andean montane forests and the Southern Atlantic Forest [43–45]. At least two corridors may have connected the Andean region and the Atlantic Forest, one through the Chaco region, passing through Paraguay, northern Argentina and Bolivia, and the other within the Cerrado region, connecting the central Atlantic Forest and the Andes by gallery forests [45].

Snakes are an interesting model for understanding biogeographical processes. The group started to radiate more than 100 mya [46], and is distributed worldwide with the exception of Antarctica [47–50]. Thus, a myriad of processes might have shaped their current diversity and geographic patterns. Despite the great endemism and diversity of this group in the Neotropical region [51], it is only recently that the biogeographic processes responsible for this diversity have started to be studied [52–55]. Snakes of the family Viperidae, also known as vipers, are distributed worldwide and comprise approximately 360 species [56]. However, about 70% of all viper species are endemic to the New World [57].

The genus *Bothrops* in the viperid subfamily Crotalinae, commonly known as lanceheads, represents one of the most species-rich genus occurring in the New World [56] and, with the exception of few species, most of them are endemic to South America. *Bothrops* species are distributed in both forest blocks and open areas over the Neotropics, making them an ideal model for studying biogeographical processes in the Neotropical region. The group counts with recently phylogenies available [57, 58], and The Reptile Database [56] currently recognizes 45 species of lanceheads. Within these species, a group of 18 species stands out, as it is composed mainly by forest lanceheads, including *Bothrops moojeni*, which occurs in gallery forests throughout the Cerrado region [59], a geographic domain with savannah-like open vegetation.

Here we explored the hypothesis that the expansion of the DODL acted as a vicariant process for vipers of the genus *Bothrops*, by analyzing the temporal range dynamics of a *Bothrops* clade comprising 18 forest species, and using models of ancestral geographic range estimation. We intended to answer the following questions: (1) where did our focal forest clade originate?; and (2) how biogeographic processes such as vicariance and dispersal shaped the current geographic distribution of those species?

## Materials and methods

### Study area and regionalization

Different regionalization schemes have been proposed for the Neotropical Region [60–63]. Terrestrial ecoregions of the world represent a regionalization scheme based on global floristic maps and vegetation types [63, 64]. We used combinations of Neotropical ecoregions *sensu* Olson et al. [63] and Dinerstein et al. [64] to define our biogeographic units. Specifically, we defined 13 biogeographic units by combining ecoregions according to the distribution patterns of the focal clade (Fig 2).

### Phylogenetic inference

To perform biogeographic analyses described below, we used the recently published molecular dataset provided by Carrasco et al. [58] to generate a dated molecular phylogeny for our focal clade. In their study, Carrasco et al. [58] generated a total evidence phylogeny of the genus *Bothrops*, with the description of a new species. Their dataset comprised only mitochondrial genes (12s, 16s, cox1, cytb and nd4), with 412 sequences. We used the R packages ape [65, 66] and seqinr [67] to retrieve the sequences from GenBank [68] using the accession numbers provided by Carrasco et al. [58]. We aligned the sequences with MAFFT version 7 online service [69–71] using default settings. The L-INS-I algorithm [72] was used to align genes 12s, 16s and cox1, and the FFT-NS-I algorithm [73] to align genes cytb and nd4. We used Gblocks 0.91b [74, 75] to remove poorly aligned positions of each sequence. We decided to allow gap positions within the final blocks in Gblocks. The sequences were then analysed and concatenated in MESQUITE 3.61 [76]. Our final sequence matrix contained 2,482 base pairs and 197 terminals. We used PartitionFinder 2.1.1 [77, 78] with a greedy algorithm and linked branch lengths to select the best partition scheme and corresponding substitution models for our gene matrix, partitioned by gene and codon position. The corrected Akaike Information Criterion (AICc) [79] was used to select the best fitted models (see S1 File for more info).

We estimated the phylogenetic relationships and divergence times using a Bayesian framework implemented in the software BEAST v2.6.4 [80, 81]. We estimated substitution rates using a relaxed uncorrelated lognormal clock [82]. We used a Birth-Death speciation model [83] as opposed to the Yule model [84], as the latter assumes zero extinction rates [85, 86] and it is commonly known that extinction has great importance in species diversification [87, 88]. We calibrated the phylogenetic relationships by using some of the age of divergence recovered

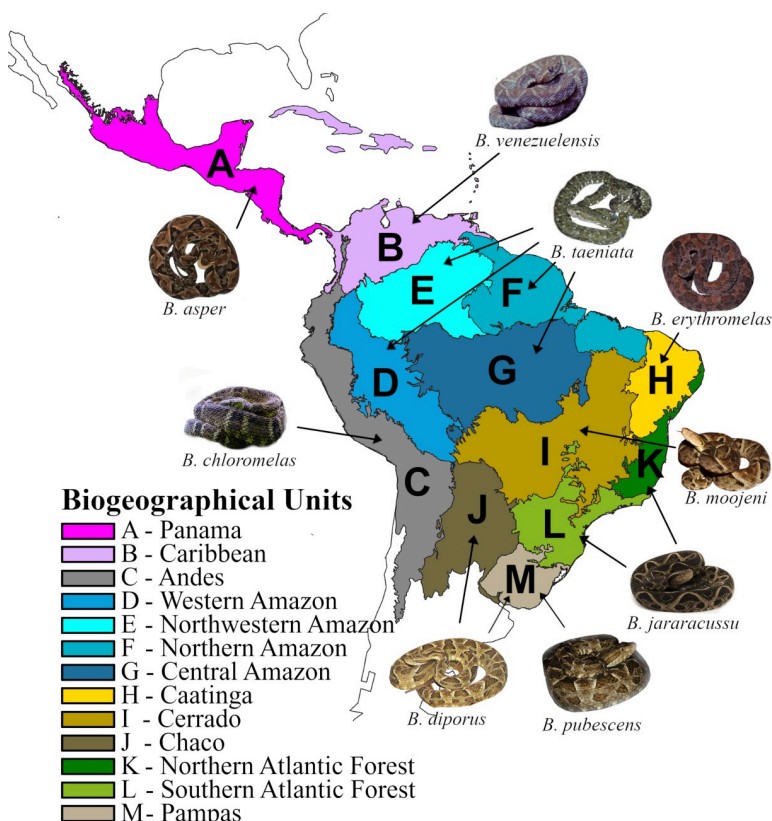

**Fig 2. Regionalization scheme used in this study.** Biogeographical units are indicated by capital letters. Ecoregions or combinations of more than one ecoregion followed Olson et al. [63] and Dinerstein et al. [64] and were based on the present geographic range of *Bothrops* species. Images of *Bothrops* species are examples of some key species in each biogeographic unit. Photograph credits: Bruno Ferreto Fiorillo (*Bothrops jararacussu*), Marcio Martins (*B. moojeni*, *B. diporus* and *B. erythromelas*), Laurie J. Vitt (*B. taeniata* and *B. venezuelensis*), Conrado Mario da Rosa (*B. pubescens*), Herpetológica LAB from Mexico (*B. asper*), and Thibaud Aronson (*B. chloromelas*). Map source: Natural Earth (www.naturalearthdata.com).

by Alencar et al. [57] (see S1 File for more detail). Alencar et al. [57] generated a dated phylogeny comprising 79% of viper species using both mitochondrial and nuclear genes (six and five, respectively). These authors also dated their phylogeny using six fossils as calibration points, being two viperid fossils and four positioned in the outgroup. Although using fossils as calibration points is the best strategy to date phylogenies [89–92], the fossil record of vipers is scarce and only few fossils are considered suitable to be used as calibration points in phylogenetic analyses [57, 91, 92]. This is even more pronounced in Neotropical vipers, such as *Bothrops*. For this reason, we decided to date our phylogeny by using dates estimated by Alencar et al. [58], which were able to include several calibration points given their broader phylogenetic context. We decided to remove *B. colombiensis* and *B. isabelae* as they are not considered as separated species in recent taxonomic arrangements [56].

## Geographic distribution data

Geographic distributions of each species were mostly based on the 4,158 raw point records obtained from Nogueira et al. [51]. For species not present in Nogueira et al. [51], we used the maps generated by Guedes et al. [93] and descriptions of the distributions provided by The Reptile Database [56]. The distribution of *Bothrops sonene* is available in Carrasco et al. [58].

Distributions can be seen in S2 File and S1 Fig. Species showing marginal geographic distributions in a biogeographic unit (< 10% of the total distribution records) were not considered as occurring in that unit (S2 File).

### Ancestral geographic range estimation

Several models have been proposed to reconstruct ancestral geographic ranges [94], such as the Dispersal-Vicariance Analysis (DIVA; [95]), the Dispersal-Extinction-Cladogenesis (DEC; [96]), and the BayArea model [97]. DIVA is a parsimony model that considers vicariance more important than dispersal, giving different costs for this process (0 for vicariance, 1 for dispersal; [95]). Also, DIVA does not consider different processes that can occur among sympatric lineages (i.e. widespread and subset sympatry; [98]). DEC is a parametric model that implements two types of sympatric processes (narrow and subset; [98]). However, it lacks the implementation of widespread vicariance [98]. The BayArea is a Bayesian model that deals better with larger numbers of areas than other models, however it does not implement vicariant processes [98]. Recent implementations allow us to compare these models in a single platform, such as the package BioGeoBEARS in R software [98, 99]. BioGeoBEARS incorporates biogeographical models in a parametric and Maximum Likelihood (ML) environment. As the DIVA and BayArea are, respectively, parsimony and Bayesian models, they are implemented in BioGeoBEARS as DIVALIKE and BAYAREALIKE. These implementations are ML versions of the originals, with the biogeographic assumptions of such models, or processes that were originally implemented [98, 99], and not parsimony nor Bayesian approaches themselves. Details on how BioGeoBEARS interprets all biogeographical processes with examples from our results can be found in S1 Table.

We compared nine Maximum Likelihood (ML) models implemented in BioGeoBEARS, all of them representing variations of the three most used ML models: DEC, DIVALIKE, and BAYAREALIKE. Models can be set to estimate a certain maximum number of units in each ancestor, called the 'maximum range size'. Changes in the maximum number of units can change the number of ranges possible in each node. The lower the maximum number of units, the lower the combinations possible. We set the maximum range size to 8, which is the number of units included in the range of *Crotalus durissus*. Note that *C. durissus* is a widespread Neotropical pit viper, showing the highest number of units within its geographical range, and for this reason we assumed that the ancestors had the potential to occur in up to 8 units. Although representing one of the major novelties of the BioGeoBEARS package [98–100], we did not include the parameter "j" (jump dispersal process) in our analyses due to recent discussions involving it [101]. Three models included a time stratified dispersal matrix (the "TS" models), that arbitrarily multiplies dispersal probabilities between two different regions based on landscape evolution of the Neotropical Region. Such probabilities range from 0 to 1, with 0 meaning that a geographic barrier prevents dispersal between two areas, and 1 meaning no dispersal limitations between two areas. Because we are testing the influence of the diagonal of open/dry landscapes, we decided to use the TS in our analysis. The explanation of how the time stratified matrix was created is available in S1 Text, and the S3 File contains the matrix itself. All models were compared using AIC [102], and the best fitting models were then analysed and discussed. As a supplementary analysis, we also performed the biogeographic reconstructions using the phylogeny generated by Alencar et al. [57] (S2 Fig).

## Results

The topology recovered in the present study is similar to recent phylogenies generated for the group (S3 Fig) [57, 58]. More specifically, we also recovered *Bothrops* and *Bothrocophias* as

**Table 1. Model comparisons on the ancestral range reconstruction of *Bothrops* forest clade performed with the phylogeny generated in this work.** *d*, *e* and *w* are free parameters in models where *d* is the rate of range expansion (i.e. dispersal), *e* is the rate of range contraction (i.e. extinction), and *w* is a dispersal multiplier parameter. As the time stratified matrix (TS models) were generated with arbitrary numbers, the "+*w*" models leave the *w* parameter free and the matrix itself is raised to the *w* parameter to seek the best dispersal multiplier values. Best models (DECTS and DIVALIKETS) are highlighted.

| Models | Log likelihood | *d* | *e* | *w* | AIC | AIC weights |
|---|---|---|---|---|---|---|
| **DECTS** | **-213.8** | **0.011** | **<0.0001** | **1** | **431.6** | **0.76** |
| **DIVALIKETS** | **-215** | **0.014** | **0.0054** | **1** | **433.9** | **0.24** |
| DIVALIKETS+w | -220.7 | 0.01 | <0.0001 | 0.014 | 447.4 | 0.0003 |
| DEC | -221.8 | 0.0078 | <0.0001 | 1 | 447.6 | 0.0003 |
| DIVALIKE | -222.1 | 0.0095 | 0.002 | 1 | 448.1 | 0.0002 |
| DECTS+w | -221.8 | 0.0078 | <0.0001 | 0.0013 | 449.6 | <0.0001 |
| BAYAREALIKETS | -248.9 | 0.014 | 0.13 | 1 | 501.7 | <0.0001 |
| BAYAREALIKE | -252.5 | 0.011 | 0.14 | 1 | 508.9 | <0.0001 |
| BAYAREALIKETS+w | -252.3 | 0.011 | 0.14 | 0.037 | 510.7 | <0.0001 |

paraphyletic due to the placement of *B. lojanus* within the latter. Within *Bothrops*, all major groups were recovered with high posterior probabilities, including the *B. atrox* (100%), *B. jararacussu* (100%), *B. jararaca* (100%), *B. alternatus* (96,64%) and *B. neuwiedii* groups (96,17%). After the removal of *B. colombiensis* and *B. isabelae*, our final phylogeny had 46 terminals.

We recovered the DECTS and DIVALIKETS models as best and second best model, respectively (Table 1). Both models included the time stratified matrix. DECTS was the best fitting model and had a better fit than the second-best evaluated model. However, the AIC difference between these two models was smaller than four, and the AIC weight of the second-best model was considerably high (0.24). Moreover, the DIVALIKETS was also recovered as the best fitted model when using the phylogeny generated by Alencar et al. [57] (S2 Fig). We therefore, consider to not have enough evidence to support one model over the other [103], and considered both models that estimated ancestral ranges under the phylogeny generated in this work to discuss the temporal range dynamics and evolutionary history of the group.

The most probable ancestral range reconstruction suggested the origin of the focal forest clade in the northwestern portion of South America (Node 1; unit D in DIVALIKETS and units ACD in DECTS, Fig 3 and S4 Fig, respectively). The Western Amazon biogeographical unit (D; see Fig 3) was recovered in both models in Node 1. However, geographical range probabilities reconstructed in DECTS are highly unresolved (S4 Fig), meaning that there are several other equally plausible combinations of biogeographical units that potentially represent the ancestral geographical range of the clade (this is why the pie chart is covered in black). Specifically, the range ACD has a probability of only 2.65% (S4 File) and is still the most probable ancestral range for this node, followed by ACDF (2.03%) and ABCDEF (1.90%). Under the DIVALIKETS model, on the other hand, ancestral range reconstruction is much more probable (Fig 4), as the range D has a probability of 33.33% (S4 File), followed by AD (17.85%) and DL (12,27%). The best-fitted model for Alencar et al.'s phylogeny [57] also recovered the Western Amazon as the most probable ancestral distribution of the forest focal clade (S2 Fig).

Vicariant processes occurred in nodes splitting lineages into two or more biogeographic units to its descendants (See Fig 3 and S4 Fig, and S1 Table for more details). In the *Bothrops jararacussu* group (Node 8), a vicariant process must have occurred splitting the lineages occurring in the Western Amazon (D) and the Southern Atlantic Forest (L) units (DL → L for the upper descendant, D for the lower descendant). Other vicariant processes must have taken place within the *B. atrox* group (Node 12). In Node 16, a vicariant event splitted lineages occurring in the Northern Amazon (F) and Northern Atlantic Forest (K) units (FK → K, F).

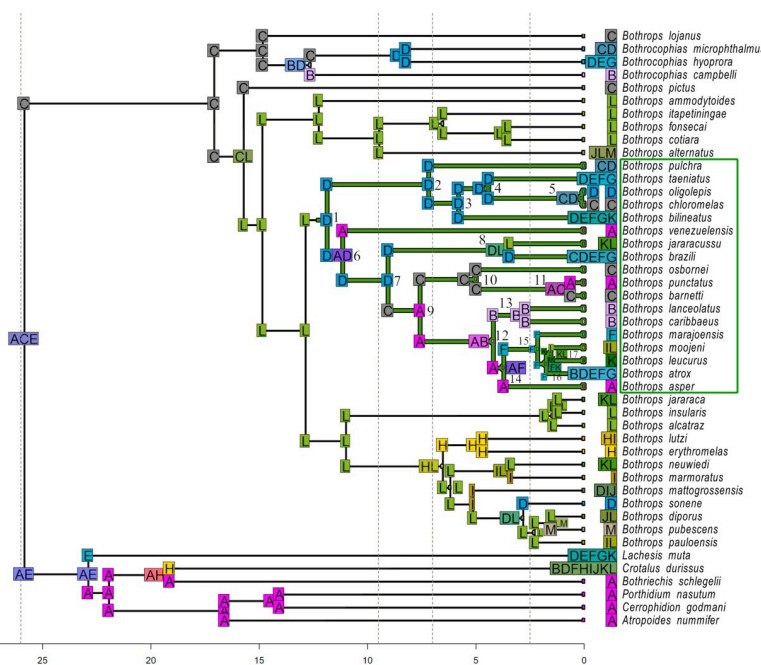

**Fig 3. Ancestral geographic ranges of the *Bothrops* forest clade reconstructed under the DIVALIKETS model.**
Single capital letters indicate different biogeographic units used in this study. Mixed letters represent combinations of such units. Colors also represent biogeographic units. Combinations of two or more units are shown as a mixed colour made from all units in the combination. Units next to species names represent the current geographic distribution of each species. The green clade highlights the focal forest clade. Vertical dashed grey lines indicate the time slices defined in the time stratified matrix (see Methods). Letters in the corners of the cladogram represent the geographical range inherited from the ancestor immediately after a cladogenetic process. Focal nodes discussed in the text are numbered.

Other vicariant processes in the forest clade explain diversification at nodes 5 (CD → D, C), 6 (AD → A, D), 14 (AF → A, F), and 17 (KL → L, K).

Dispersal processes (see S1 Table for details) between the Amazon and Atlantic forests may have primarily occurred: (i) before the ancestor of the *B. jararacussu* group emerged, between Node 7 upper descendant and Node 8 (D → DL); (ii) within the *B. atrox* group, between Node 15 lower descendant and Node 16 (F → FK); and in the lineage giving rise to *B. bilineatus* (D → DEFGK). Other dispersal events occurred in the lineage giving rise to *B. moojeni*, the only species inside the focal forest clade that reached the DODL, more specifically the Cerrado (L → IL).

## Discussion

In this study, we investigated how the geographic ranges of a clade comprising 18 forest lanceheads changed across time, and which biogeographic processes were involved during the diversification of the clade in the Neotropical Region. The Western Amazon (D) biogeographic unit was the most probable ancestral geographic range of the clade, both for models selected using our phylogeny, based on the molecular dataset provided by Carrasco et al. [58] (Fig 3 and S4 Fig), and the best fitted model using Alencar et al.'s phylogeny [57] (S2 Fig). However, this result remains uncertain under the DECTS model (black pie charts in S4 Fig). Despite the DIVALIKETS being much more resolved, the Panama unit also appears at this node with high probability, representing the second most probable range distribution for the ancestor of the clade under DIVALIKETS. If a region comprising both Western Amazon and

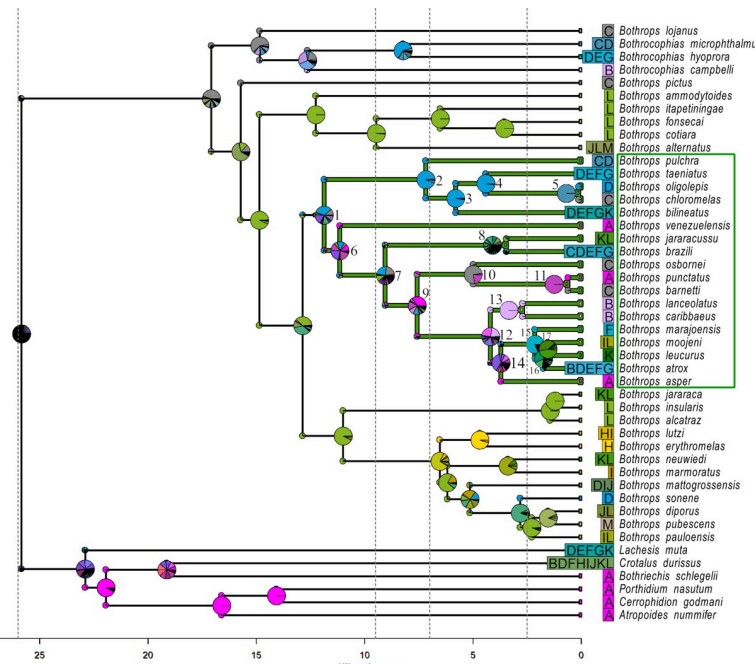

**Fig 4. Ancestral geographic range probabilities associated with the range reconstruction of *Bothrops* forest clade under the DIVALIKETS model.** Pie charts in nodes represent the geographic range probabilities of each hypothetical ancestor. Pie charts in the corners of the cladogram represent the geographical range probabilities inherited from the hypothetical ancestor immediately after a cladogenetic process. Colors represent biogeographical units. Units next to species names represent the current geographic distribution of each species. The green clade highlights the focal forest clade. Vertical dashed grey lines mark the time slices defined in the time stratified matrix. Focal nodes discussed in the text are numbered.

Panama units were the range of the clade's ancestor, a vicariant process would have happened at the onset of the diversification of the clade. This vicariant event could have been potentially related to changes in the landscape that were occurring in those regions at the time, such as the uplift of the Andes [19–21] and the sea introgressions forming the Pebas system [21, 38, 104, 105]. Nevertheless, the most probable combinations of ancestral biogeographic units point to a northwestern South America origin for the forest clade, and most of these combinations include Amazonian units.

When focusing only on the groups that have a disjunct distribution on the forest blocks of the Neotropical Region, that is, those splitted by the DODL, a pattern of dispersal events followed by vicariance was detected, specifically vicariance events at nodes 8 and 16 and previous dispersal events. These events agree with the hypothesis of the expansion of the DODL in the Neotropical Region acting as a vicariant process, as already suggested for different groups of organisms [24, 30]. The formation of the diagonal was gradual and followed the late uplift of the Andes [14, 21], that reached its present height from 6 to 4 million years ago–mya [22]. This pattern of dispersal and vicariance is present in estimates under the DECTS (S4 Fig), DIVALIKETS (Figs 3 and 4) and under the best-fitted model (DIVALIKETS) using the phylogeny generated by Alencar et al. [57] (S2 Fig).

Fouquet et al. [28] found a similar pattern for the frog genus *Adenomera*, with a possible direct dispersal from the Amazon Forest to the Atlantic Forest followed by a vicariant process. According to those authors, the timeframe of this dispersal is concomitant with the ancient continuity between the two forest blocks [28]. It is probable that the formation of the DODL and subsequent split of the ancient forest connection led to the patterns observed. Other

studies found dispersal processes between the two forest blocks followed by cladogenetic processes and lineage isolation. For the lizards of the genus *Enyalius*, the colonization of the Amazon from an Atlantic Forest ancestor occurred within the Oligocene (~ 25 mya) and led to further isolation of *Enyalius leechii* [27]. A similar pattern was observed for the lizards of the genus *Anolis*, although in the opposite direction, the colonization of the Atlantic Forest by an Amazonian ancestor [42].

Both vicariant processes found in our study (nodes 8 and 16) must have occurred in Plio-Pleistocene epochs (3–2 mya), which occurred more recently than those found by Fouquet et al. [28]. Some authors suggested much older vicariant events, between the Oligocene [27, 33] and early Miocene [28, 42]. Simon et al. [106] discuss that much of the Cerrado plant lineages began to diversify in the late Miocene (10 mya), and this late grassland expansion agrees with the even later vicariant events that we found here for the *Bothrops* clade. Coscarón and Morrone [24, 30], analysing members of the bug family Peiratinae, discuss that the disjunction patterns between the Atlantic and Amazon forests observed in many organisms occurred as a result of vicariant events, probably related to the expansion of the DODL influenced by the late Andean uplift in the Plio-Pleistocene, inducing aridity in South America through the interruption of the Pacific airflow [24, 107]. The late uplift of the Andes is also linked to the subsidence of the Chaco and the uplift of the Brazilian Plateau [13, 23]. Martins et al. [108] also found that the emergence of the DODL is congruent with a vicariant process in populations of the vampire bat *Desmodus rotundus* during the Pleistocene. A similar pattern of pleistocenic vicariance was suggested for birds from the genus *Xiphorhynchus* [109]. Forest expansions and contractions occurred at different times along the DODL [9, 12, 28, 106, 107], and this could explain the effect of the DODL in different timeframes. The results found here are congruent to these climatic and geological modifications in South America.

The only dispersal between the two forest blocks not followed by a vicariant event occured in the lineage giving rise to *B. bilineatus*. This species currently has a disjunct distribution in both the Atlantic and Amazon forests. Dal Vechio et al. [110] concluded in a phylogeographic approach that two different dispersal events probably occurred for this species, one at 2 mya ago, when the Atlantic Forest was colonized by a western Amazonian ancestor, and one more recently, at 0.3 mya, when an Atlantic Forest ancestor dispersed back to the Amazon Forest through the northeastern coast of Brazil [110]. The second dispersal event would be in line with the forest expansions and climatic fluctuations that occurred during the Quartenary [9, 11, 12], through the "young pathways" between the Atlantic and Amazon forests, as classified by Batalha-Filho et al. [34]. However, these pathways might have occurred much later [9, 11] and would correspond to the date of the second dispersal by *B. bilineatus*.

Perhaps more important than the vicariant processes themselves that occurred during the diversification of the clade of forest lanceheads is the almost lack of dispersal processes between the Amazon and the Atlantic forests. Dispersal events took place mostly: (i) within the Amazon units or between these Amazon units and the Caribbean and/or Andean units (B, C, D, E, F and G); and (ii) within the two Atlantic Forest units (K and L). Exceptions to this pattern are the three dispersals discussed above. This reinforces the DODL as a dispersal barrier between the two forest blocks, limiting the dispersal between them even after the Plio-Pleistocene climate fluctuations (2.5 mya) and forest expansions within the diagonal [9, 12, 16, 31]. Similar results are found for the frog genus *Dendrophryniscus* and *Amazonella* [33], *Adenomera* [28], and for the lizard genus *Leposoma* [29]. The restriction to forests and the lack of dispersion through open areas by forest lanceheads might suggest that niche conservatism could be prevalent in this clade. Indeed, the few dispersal events between the Amazon and Atlantic forests may even have happened through the forest corridors present within the DODL in the past, as highlighted for *B. bilineatus* [110]. Future studies could shed light on the

usage of these corridors by other members of the *Bothrops* genus. Regardless of the pathways between the Amazon and the Atlantic forests within the DODL, our study indicates that the upper Miocene and Pliocene history of our clade was heavily influenced by the emergence of the DODL, as it may have acted both as a vicariant process in some lineages and as a dispersal barrier within our focal clade.

## Conclusions

The ancestor of our focal clade of forest lanceheads was distributed in the northwestern Amazon forests, and the clade diversified over the Neotropical Region. Most of the dispersal events occurred within the Amazon and the Atlantic Forest, and not between them. Then, the DODL may have acted as a dispersal barrier between those forests. Moreover, the expansion of the diagonal is likely to have acted as a vicariant process for two clades of forest lanceheads.

## Supporting information

**S1 File. Beauti configurations.** The file consists of three sheets, the first showing the partitions used, the second the priors (without the calibration points) and the third the calibration points.
(XLSX)

**S2 File. Summary of the species distributions.** The file consists of two sheets. The first shows a summary of the distributions of all species, including source, range distribution, distributions not considered and reasons for not considering them. The source has a reference, and the number on its right side corresponds to the reference number present on References. The second sheet consists of the geographical file needed to run BioGeoBEARS. The numbers on the first line correspond to the number of species and biogeographical units used in this study, respectively. The letters correspond to the units. The 1s and 0s indicate the presence or absence of the species in an area, respectively. The order of the numbers corresponds to the order of the units in the first line.
(XLSX)

**S3 File. The time stratified matrix file.** It is divided into time slices corresponding to windows of millions of years. The values inside the matrix multiplies the dispersal probabilities between two areas (i.e. units), where 1 means no influence in the dispersal probability (total possibility of dispersal between two areas) and 0 means total influence in the dispersal probability (no possibility of dispersal between two areas). The values are all arbitrary. However, they are based on the known landscape evolution of the region covered by this study. Each line and column represent a unit. The units represented in the lines are the units from where the species dispersed, and in the columns are the units to which the species dispersed.
(TXT)

**S4 File. Percentages of every combination of units possible for every node in the forest lanceheads clade.** The nodes correspond to those present in the graphical results of both models (Figs 3 and 4 and S4 Fig). The file consists of two sheets, the first being for the DECTS model and the second being for the DIVALIKETS model.
(XLSX)

**S1 Fig. Distribution map of the species utilized in this study.** The occurrences indicated by squares are from Guedes et al. [93], and those dots are from Nogueira et al. [51]. The distributions that were recovered from Uetz et al. [56] and Carrasco et al. [58] are not shown, as they are descriptions. The units correspond to the units used in this study (Fig 2). Map made with

Natural Earth. Free vector and raster map data from naturalearthdata.com.
(TIFF)

**S2 Fig. Most probable ancestral range reconstructed by DIVALIKETS using Alencar et al.'**
**phylogeny.** Single capital letters indicate different biogeographical units used in this study.
Mixed letters represent combinations of units. Colours also represent biogeographical units.
Combinations of two or more units are shown as a mixed colour made from all the units in the
combination. Units next to species names represent the current geographical distribution of
each species. The green clade showcases the focal forest clade. Vertical dashed gray lines mark
the time slices defined in the time stratified matrix. Letters in corners of the cladogram repre-
sent the geographical range inherited from the ancestor immediately after a cladogenetic pro-
cess.
(TIFF)

**S3 Fig. Maximum credibility tree generated with Carrasco et al. [58] gene dataset used in**
**this study.** Green clades were collapsed in the final phylogeny. Red clades were removed from
the final phylogeny. Posterior probabilities higher than 0.75 are present at nodes (for some
internal nodes we also occulted some posteriors for better visualization).
(TIFF)

**S4 Fig. Graphical results of DECTS.** (A) Ancestral geographic ranges reconstructed by the
model. (B) Ancestral geographic range probabilities reconstructed by model. Nodes from the
forest lanceheads clade are labelled. Single capital letters indicate different biogeographic units
used in this study. Mixed letters represent combinations of units.Units next to species names
represent the current geographic range of each species. The green clade showcases the focal
forest clade. Vertical dashed grey lines mark the time slices defined in the time stratified
matrix. Letters in corners of the cladogram represent the geographic range inherited from the
ancestor immediately after a cladogenetic process.
(TIFF)

**S1 Table. Description of the processes considered by the models used in the work.** Exam-
ples are from the DIVALIKETS model. The table was heavily inspired and based on Matzke
[98].
(DOCX)

**S1 Text. Explanation on how the time stratified matrix file was built.**
(DOCX)

## Acknowledgments

We thank M. Carolina R. Manzano, Marcela Brasil Godinho, Leonardo M. Servino, Bruna
Bolochio, and the Laboratório de Evolução e Diversidade 1 (LED1-UFABC) for insights and
inspirations in earlier versions of the manuscript. We also thank Sam Hardman for English
revision.

## Author Contributions

**Conceptualization:** Matheus Pontes-Nogueira, Marcio Martins, Ricardo J. Sawaya.

**Data curation:** Matheus Pontes-Nogueira, Ricardo J. Sawaya.

**Formal analysis:** Matheus Pontes-Nogueira, Laura R. V. Alencar.

**Funding acquisition:** Ricardo J. Sawaya.

**Investigation:** Marcio Martins, Laura R. V. Alencar.

**Methodology:** Matheus Pontes-Nogueira, Laura R. V. Alencar.

**Project administration:** Marcio Martins, Ricardo J. Sawaya.

**Resources:** Marcio Martins, Ricardo J. Sawaya.

**Software:** Laura R. V. Alencar.

**Supervision:** Laura R. V. Alencar, Ricardo J. Sawaya.

**Validation:** Ricardo J. Sawaya.

**Visualization:** Matheus Pontes-Nogueira.

**Writing – original draft:** Matheus Pontes-Nogueira.

**Writing – review & editing:** Matheus Pontes-Nogueira, Marcio Martins, Laura R. V. Alencar, Ricardo J. Sawaya.

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
