## [Decision Letter · Decision Letter 0]

13 May 2021

PONE-D-21-11305

The role of vicariance and dispersal on the evolution of geographic distribution of forest vipers in the Neotropical region

PLOS ONE

Dear Dr. Pontes-Nogueira,

Thank you for submitting your manuscript to PLOS ONE. After careful consideration, we feel that it has merit but does not fully meet PLOS ONE’s publication criteria as it currently stands. Therefore, we invite you to submit a revised version of the manuscript that addresses the points raised during the review process.

We look forward to receiving your revised manuscript.

Kind regards,

Tzen-Yuh Chiang

Academic Editor

PLOS ONE

Journal Requirements:

2.We note that Figure(s) 1 in your submission contain map images which may be copyrighted. All PLOS content is published under the Creative Commons Attribution License (CC BY 4.0), which means that the manuscript, images, and Supporting Information files will be freely available online, and any third party is permitted to access, download, copy, distribute, and use these materials in any way, even commercially, with proper attribution. For these reasons, we cannot publish previously copyrighted maps or satellite images created using proprietary data, such as Google software (Google Maps, Street View, and Earth). For more information, see our copyright guidelines: http://journals.plos.org/plosone/s/licenses-and-copyright.

a)  You may seek permission from the original copyright holder of Figure(s) 1 to publish the content specifically under the CC BY 4.0 license. 

Reviewers' comments:

Reviewer's Responses to Questions

**Comments to the Author**

1. Is the manuscript technically sound, and do the data support the conclusions?

Reviewer #1: No

Reviewer #2: Yes

2. Has the statistical analysis been performed appropriately and rigorously? 

Reviewer #1: No

Reviewer #2: Yes

3. Have the authors made all data underlying the findings in their manuscript fully available?

Reviewer #1: Yes

Reviewer #2: No

4. Is the manuscript presented in an intelligible fashion and written in standard English?

Reviewer #1: Yes

Reviewer #2: Yes

5. Review Comments to the Author

Reviewer #1: Comments to authors:

Your manuscript entitled” The role of vicariance and dispersal on the evolution of geographic distribution of forest vipers in the Neotropical region" deals with an interesting topic. Unfortunately, however, I must reject your manuscript in the current version for the following reasons:

(1) I believe you need to reconstruct a new phylogenetic tree and divergence time tree comprising all species of the genus Bothrops. I suggest you generate a new dataset for all the five mtDNA genes which are common between Alencer et al.’s (2016) and Carrasco et al.’s (2019) studies. Alencer et al.’s dataset (2016) had numerous gaps and some species were partially covered for the 11 genes. For instance, Bothrops alcatraz was only covered for one gene (cyt b). For this reason, the divergence time tree is not reliable for BioGeoBEARS analysis, as it is sensitive to branch lengths.

(2) It appears that the results of your BioGeoBEARS analysis are not robust as you disregarded the four new species of the genus Bothrops.

Further Details regarding your BioGeoBEARS analysis:

- Line 164: What was the age of the root node? Was it included in the time slices?

- Line 164: What was the maximum range used for the BioGeoBEARS?

- Line 164: I would recommend considering the Area-adjacency, Area-allowed, and distance matrix in the analysis.

Reviewer #2: In this work, Pontes-Nogueira and collaborators address the macro-evolution of Bothrops pitvipers and their temporal range dynamics in South America, using for such purpose ancestral range reconstructions based on phylogenetic comparative models. Specifically, they test if the expansion of arid environments affected the evolution of this group. Despite the interest of the work, the current version of the ms requires to be improved for clarity in several aspects and along the distinct sections. Approaches and decisions have to be better explained and justified. The graphical part is quite limited and requires to be improved; and the discussion is a bit an extension of the results, without considering other terrestrial fauna that evolved in the region. Below I provide some comments to help the authors improve the ms.

Abstract

Line 19: terrestrial biodiversity rather than some lineages

Line 23: temporal range dynamics?

Lines 33-35: this conclusion is repetitive. Perhaps you can highlight how your work contribute to a better understanding of the biogeography and evolution of terrestrial biodiversity in South America.

Introduction

Lines 37 – 38: this first sentence is not well linked to the following information. Please consider to remove it, you can start with the second sentence in a very good way.

Line 46-48: do these vicariant processes result from the Pleistocene climatic oscillations (lines 44-46)? Pleistocene is a rather modern period and speciation events in reptiles (i.e. strictly referring to the formation of species) usually predate this time. Information from lines 45 to 48 must be set in the context of your work. In fact, in the next paragraph you explain the expansion of the DDL since the Oligocene and therefore, information in lines 45 to 48 seems out of context.

Line 49: Is it the expansion of the DDL or the existence/occurrence of these landscapes in between other?

Lines 49-104: isn’t any way to schematically represent these scenarios in a figure to improve the understanding of these processes and the location of barriers/corridors?

Line 64-65: I think it is better to refer to “terrestrial biodiversity” or something similar than “some lineages”

Lines 105-106: please, state why snakes are such interesting model.

Lines 111-113: please provide more information about this group of vipers. N of spp, ecology, habitat, ... etc.

Line 117: information about this forest clade should be clearly stated in the introduction. Is it constituted by the whole Bothrops? Or just by some lineages?

Line 118: please reinforce the biogeographical processes that you are thinking

Material & methods

Lines 125-126: why didn't you use the most recent work of these authors about biogeographical units?

Dinerstein, E., Olson, D., Joshi, A., Vynne, C., Burgess, N. D., Wikramanayake, E., ... & Saleem, M. (2017). An ecoregion-based approach to protecting half the terrestrial realm. BioScience, 67(6), 534-545.

Lines 142-146: the phylogeny of Bothrops with the supporting nodes could be presented in figure 1, with the map, specifying the major habitats for the clades (forest...).

Lines 151-155: this decision has to be better grounded. If the new phylogeny has more information that the one you are using but it is not dated, you can either (1) reconstruct a newer phylogeny for the group considering the information in both phylogenies and then date it; or (2) use the dating in Alencar et al to calibrate the new (and more complete) phylogeny.

Lines 158-159: a spatial representation of the distribution will help the reader to understand the following approach

Line 161: this explanation is vague. You could provide more info about what you consider "biological factors" here, in the main text.

Lines 165-184: I am missing some information about the models itself, what are their differences? Just provide a brief text for each one.

Lines 179-181: time stratified dispersal matrix could be provided in SM

Results

Is not there any way to graphically summarise range dynamics of Bothrops according to your results?

Lines 190-192: perhaps this information could be available in numeric format in SM

Lines 194-197: this interpretation of the results fits better in the discussion.

Line 242: please, explain how you reach this result. While ancestral ranges is easy to understand, I see complicated to follow your results for dispersal. This information is needed in M&M.

Lines 227-248: this text is quite difficult to follow because the figures have no numbers in the nodes

Discussion - it repeats the results and rarely goes far from the Bothrops group. Are not other groups that could support the discussion of your findings? In the introduction, several examples are provided to support the DDL. How do the studies developed in these fauna relate to your findings?

Line 251: first time using lanceheads!! If you use this term, this must be referred in the introduction.

Lines 262-263: please explain this better. What do you refer with “inclusive”?

Lines 308-310: which kind of recent studies? Please provide more information

Lines 314-315: please provide some examples of these future studies, will they target on genomics, landscape analysis, …?

Lines 315-318: this sentence fits much better in the conclusion

Conclusions

Briefly reinforce how by addressing the DDL hypothesis in Bothrops, your work does contribute to a better understanding of the biogeography and evolution of terrestrial biodiversity in South America.

Figure 1 – please consider to include the letters for each biogeographical unit (in the map) in the legend. Also, you could include the phylogeny, clearly signalling which the focal forest clade is.

6. PLOS authors have the option to publish the peer review history of their article (what does this mean?). If published, this will include your full peer review and any attached files.

Reviewer #1: No

Reviewer #2: No

---

## [Author Response · Author response to Decision Letter 0]

26 Jul 2021

We are very thankful for the comments and revisions on our manuscript made by the academic editor and the two reviewers of the initial text. Below is our response to every single point raised by both the editor and the reviewers to improve our work and to fully meet PLOS ONE’s publication criteria. We hope that we satisfyingly addressed these points and that our manuscript will now be suited for publication. 

Sincerely,

On behalf of all authors,

Matheus Pontes-Nogueira (corresponding author).

Academic Editor

Answer: File naming was changed to comply with the PLOS ONE’s style requirements. We hope that now names are in line with style requirements now.

2.We note that Figure(s) 1 in your submission contain map images which may be copyrighted. All PLOS content is published under the Creative Commons Attribution License (CC BY 4.0), which means that the manuscript, images, and Supporting Information files will be freely available online, and any third party is permitted to access, download, copy, distribute, and use these materials in any way, even commercially, with proper attribution. For these reasons, we cannot publish previously copyrighted maps or satellite images created using proprietary data, such as Google software (Google Maps, Street View, and Earth). For more information, see our copyright guidelines: http://journals.plos.org/plosone/s/licenses-and-copyright.

a) You may seek permission from the original copyright holder of Figure(s) 1 to publish the content specifically under the CC BY 4.0 license. 

Answer: We really appreciated this suggestion and opted to follow the second path. We remade all maps present in our work with public domain maps made available by Natural Earth. All map captions now have a disclaimer about the map construction, with a “Map source: Natural Earth (www.naturalearthdata.com).” at the end. We hope that now our images will not have any copyright implications. 

Reviewer #1 

Your manuscript entitled” The role of vicariance and dispersal on the evolution of geographic distribution of forest vipers in the Neotropical region" deals with an interesting topic. Unfortunately, however, I must reject your manuscript in the current version for the following reasons:

(1) I believe you need to reconstruct a new phylogenetic tree and divergence time tree comprising all species of the genus Bothrops. I suggest you generate a new dataset for all the five mtDNA genes which are common between Alencer et al.’s (2016) and Carrasco et al.’s (2019) studies. Alencer et al.’s dataset (2016) had numerous gaps and some species were partially covered for the 11 genes. For instance, Bothrops alcatraz was only covered for one gene (cyt b). For this reason, the divergence time tree is not reliable for BioGeoBEARS analysis, as it is sensitive to branch lengths.

Answer: we thank you immensely for this comment, as it was really helpful to fully improve our work. Both reviewers critiqued that we decided to not use the Carrasco et al. (2019) dataset, so we generated a new phylogeny using this dataset under a Bayesian framework. We then used calibration points accordingly with the dates estimated by Alencar et al. (2016) to date our phylogeny, as stated in the Material and Methods section of the new improved manuscript (lines 151 – 186, revised manuscript). However, we decided to not join both datasets and opted to run two separated analyses, one in the new phylogeny generated with the Carrasco et al.’s dataset and one with the phylogeny generated by Alencar et al. We then discuss the results in light of the results of both phylogenies, that are present both as figures in our manuscript and supplementary materials from our work. To help us with this work, we decided to include as an author Drª. Laura R. V. Alencar, the researcher responsible for the Alencar et al.’s phylogeny. We hope that this decision satisfyingly addresses this point raised by both reviewers. 

(2) It appears that the results of your BioGeoBEARS analysis are not robust as you disregarded the four new species of the genus Bothrops.

Answer: Thanks for this advice. As stated before, we opted to construct a phylogeny using Carrasco et al.’s dataset, and we hope that now our analysis is robust with the inclusion of new species. However, it is worth noticing that the genetic dataset presented by Carrasco et al. do not include some of the species that these authors used to generate their phylogeny (that being B. sanctaecrusis, B. roedingeri, B. pirajai, B. otavioi and B. jonathani). This is due to lack of available genetic information of these species. Carrasco et al. generated a phylogeny using total evidence, and for these species they included morphological information that we can not use to generate a dated phylogeny with the methods used here. However, every specie that they did include a GenBank voucher we certainly used in our phylogenetic analysis.

Further Details regarding your BioGeoBEARS analysis:

- Line 164: What was the age of the root node? Was it included in the time slices?

Answer: the age of the root node was approximately 14 mya, as stated in our old supplementary material (titled S4 text in the old manuscript; line 25, fifth paragraph). As the age of the root node could be graphically seen in the results of the models, we decided to omit this information from the text. And indeed, it was included in the times slices (also stated in this supplementary material).

- Line 164: What was the maximum range used for the BioGeoBEARS?

Answer: previously it was set to 5. However it was set to 8 now due to the distribution of Crotalus durissus. We added a note in the Materials and Methods (lines 219 – 220) in the revised manuscript. Thanks for the suggestion.

- Line 164: I would recommend considering the Area-adjacency, Area-allowed, and distance matrix in the analysis.

Answer: Thanks for the suggestion. These models are very important, however our work is interest in the impact of the open/dry diagonal in the South America and, for this, we opted for not use these variations, as the time-stratified matrix already encompasses these landscape evolutions. 

Reviewer #2

In this work, Pontes-Nogueira and collaborators address the macro-evolution of Bothrops pitvipers and their temporal range dynamics in South America, using for such purpose ancestral range reconstructions based on phylogenetic comparative models. Specifically, they test if the expansion of arid environments affected the evolution of this group. Despite the interest of the work, the current version of the ms requires to be improved for clarity in several aspects and along the distinct sections. Approaches and decisions have to be better explained and justified. The graphical part is quite limited and requires to be improved; and the discussion is a bit an extension of the results, without considering other terrestrial fauna that evolved in the region. Below I provide some comments to help the authors improve the ms.

Answer: We thank the reviewer for considering our manuscript of great interest. We are also very thankful for your considerations regarding our manuscript. We took into account almost all your considerations and indeed we all agree that our work is much more improved now. Specifically, we provide more details regarding our approaches and decisions, modified some aspects of our figures and included a new one (see Fig 1 in the new version of the manuscript). We also modified several aspects of the supplementary materials and made substantial changes in our results and discussion, providing more examples regarding the biogeography of other terrestrial organisms in the region. All of these aspects are detailed below in a our answers for each specific consideration.

Abstract

Line 19: terrestrial biodiversity rather than some lineages

Answer: Agreed! We changed this in the new revised manuscript. Thanks.

Line 23: temporal range dynamics?

Answer: We changed this in the new revised manuscript. We also changed our title to encompasses this suggestion. Thanks a lot.

Lines 33-35: this conclusion is repetitive. Perhaps you can highlight how your work contribute to a better understanding of the biogeography and evolution of terrestrial biodiversity in South America.

Answer: we changed this in the revised manuscript, highlighting the importance of the open/dry diagonal in shaping distributions in South America. Thanks.

Introduction

Lines 37 – 38: this first sentence is not well linked to the following information. Please consider to remove it, you can start with the second sentence in a very good way.

Answer: Agreed! We removed this line in revised manuscript. Thanks.

Line 46-48: do these vicariant processes result from the Pleistocene climatic oscillations (lines 44-46)? Pleistocene is a rather modern period and speciation events in reptiles (i.e. strictly referring to the formation of species) usually predate this time. Information from lines 45 to 48 must be set in the context of your work. In fact, in the next paragraph you explain the expansion of the DDL since the Oligocene and therefore, information in lines 45 to 48 seems out of context.

Answer: Thanks for this comment. We reorganized this paragraph to make it better to understand and decided to remove the phrase that seems out of context.

Line 49: Is it the expansion of the DDL or the existence/occurrence of these landscapes in between other?

Answer: the expansion, that would be correlated with a vicariant processes accordingly with the hypothesis tested in the work!

Lines 49-104: isn’t any way to schematically represent these scenarios in a figure to improve the understanding of these processes and the location of barriers/corridors?

Answer: we are very thankful for this important comment, as it lead us to develop a figure that schematically represents all of these scenarios and times. The new figure (Fig 1 in the revised manuscript) graphically summarises all the landscape changes in South America regarding the expansion of the DODL. We believe that this image can help the reader to understand these changes in South America much more than the text (although the text is indispensable for explaining these changes as well). Once again we thank your for this suggestion.

Line 64-65: I think it is better to refer to “terrestrial biodiversity” or something similar than “some lineages”

Answer: Agreed! We changed this in revised manuscript. Thanks.

Lines 105-106: please, state why snakes are such interesting model.

Answer: we included a statement to why these organisms are so importante for biogeographic analysis (lines 108 – 111 in the revised manuscript). Thanks.

Lines 111-113: please provide more information about this group of vipers. N of spp, ecology, habitat, ... etc.

Answer: We provided much more information about this group in the revised manuscript (lines 116 – 125 in the revised manuscript). Thanks! 

Line 117: information about this forest clade should be clearly stated in the introduction. Is it constituted by the whole Bothrops? Or just by some lineages?

Answer: We provided much more information about this specific clade (lines 122 – 125 in the revised manuscript). It is constituted by a clade of 18 species from the Bothrops genus. Thanks for this comment.

Line 118: please reinforce the biogeographical processes that you are thinking

Answer: Thanks. We reinforced the biogeographical processes (line 130).

Material & methods

Lines 125-126: why didn't you use the most recent work of these authors about biogeographical units?

Dinerstein, E., Olson, D., Joshi, A., Vynne, C., Burgess, N. D., Wikramanayake, E., ... & Saleem, M. (2017). An ecoregion-based approach to protecting half the terrestrial realm. BioScience, 67(6), 534-545.

Answer: Thanks for the suggestion. We were unaware about this recent work and considered it in our revised manuscript. However, it is worth noticing that almost nothing has changed for the Neotropical Region (i.e. the study region of the work) in this new work compared with the older work from Olson et al. So, we decided to cite both works in our regionalization scheme. 

Lines 142-146: the phylogeny of Bothrops with the supporting nodes could be presented in figure 1, with the map, specifying the major habitats for the clades (forest...). 

Answer: thanks for this suggestion, but we preferred to stay with the regionalization image that we presented in the old manuscript (with modifications, of course) and decided to include the phylogeny as a supplementary material, as the phylogenetic inference is not the focus of this work. Regarding the major habitats, the reconstruction results show the present distribution of the species utilized in this work right before the species names. Despite it, we decided to generate a image present in the supplementary material to help the reader in knowing the distribution of the species. See the next two comments. Thanks.

Lines 151-155: this decision has to be better grounded. If the new phylogeny has more information that the one you are using but it is not dated, you can either (1) reconstruct a newer phylogeny for the group considering the information in both phylogenies and then date it; or (2) use the dating in Alencar et al to calibrate the new (and more complete) phylogeny.

Answer: we REALLY appreciate this comment. Both reviewers critiqued that we decided to not use the Carrasco et al. (2019) dataset, so we generated a new phylogeny using this dataset under a Bayesian framework. We then used calibration points accordingly with the dates estimated by Alencar et al. (2016) to date our phylogeny, as stated in the Material and Methods section of the new improved manuscript (lines 151 – 186, revised manuscript). However, we decided to not join both datasets and opted to run two separated analyses, one in the new phylogeny generated with the Carrasco et al.’s dataset and one with the phylogeny generated by Alencar et al. We then discuss the results in light of the results of both phylogenies, that are present both as figures in our manuscript and supplementary materials from our work. To help us with this work, we decided to include as an author Drª. Laura R. V. Alencar, the researcher responsible for the Alencar et al.’s phylogeny. We hope that this decision satisfyingly addresses this point raised by both reviewers.

Lines 158-159: a spatial representation of the distribution will help the reader to understand the following approach

Answer: Thanks for the suggestion. We decided to make an image with all the distribution points of the species utilized in this work to help the reader in knowing the distributions of the species. The figure is present at the supplementary materials (S3 Fig). We really appreciated this comment. 

Line 161: this explanation is vague. You could provide more info about what you consider "biological factors" here, in the main text.

Answer: we also deemed this statement confusing and decided to remove it from the manuscript. Thanks for the comment.

Lines 165-184: I am missing some information about the models itself, what are their differences? Just provide a brief text for each one.

Answer: information about the models were included in the revised manuscript for better understanding (lines 200 – 207). Thanks.

Lines 179-181: time stratified dispersal matrix could be provided in SM

Answer: we thank you for this suggestion, but the dispersal matrix was already present in the old manuscript. It was the S4 text and S3 file, and now they are the S5 text and S6 file respectively in the revised manuscript.

Results

Is not there any way to graphically summarise range dynamics of Bothrops according to your results?

Answer: thanks for the suggestion. We did not find a way to better represent these ranges dynamics besides the reconstructions per se. We elaborated further the results for better understanding and hope that it is now sufficient to understand the diversification within Bothrops. 

Lines 190-192: perhaps this information could be available in numeric format in SM

Answer: thanks a lot for this suggestion. Percentages of all range combinations at every node inside our focal clade can now be seen at the S10 File in our supplementary material. We also elaborated further the results to include the percentages in the text. Thanks once again. 

Lines 194-197: this interpretation of the results fits better in the discussion.

Answer: we agreed. We remove it from the Results. Thanks.

Line 242: please, explain how you reach this result. While ancestral ranges is easy to understand, I see complicated to follow your results for dispersal. This information is needed in M&M.

Answer: thanks for this comment. For better understanding of the process’s assumptions, we decided to create a table of processes considered in the BioGeoBEARS models. It can be seen at S4 Table in the supplementary materials of our work. We also elaborated further this paragraph for better understanding. We hope that now the assumptions about the processes will be clear to every reader.

Lines 227-248: this text is quite difficult to follow because the figures have no numbers in the nodes

Answer: Thanks for this comment. We included node numbers in our figure inside our focal clade for better understanding. Hope now it is easier to follow the results.

Discussion

 it repeats the results and rarely goes far from the Bothrops group. Are not other groups that could support the discussion of your findings? In the introduction, several examples are provided to support the DDL. How do the studies developed in these fauna relate to your findings?

Answer: Thanks a lot for this comment. We elaborated further our discussion and revised the literature about this topic to compare our results with other researchers. Most of them where cited in the Introduction and now are discussed. We hope that now our discussion meets the expectations of the reviewers! Thanks once again.

Line 251: first time using lanceheads!! If you use this term, this must be referred in the introduction.

Answer: In the revised manuscript we decided to use this term since the beginning of the manuscript (Abstract, Introduction, etc.). Thanks for the suggestion.

Lines 262-263: please explain this better. What do you refer with “inclusive”?

Answer: we agreed that this sentence was confusing and decided to remove it in the revised manuscript. Thanks.

Lines 308-310: which kind of recent studies? Please provide more information

Answer: we provided much more information about the study that we cited in this line. We discuss and elaborate further this topics in lines 376 – 385 in the revised manuscript. Thanks for the suggestion.

Lines 314-315: please provide some examples of these future studies, will they target on genomics, landscape analysis, …?

Answer: future studies could target these forest corridors cited in the work and the use of these corridors by other members of the Bothrops genus. We elaborated it further in the Discussion (lines 399 – 403 in the revised manuscript).

Lines 315-318: this sentence fits much better in the conclusion

Answer: thanks for the suggestion, but as the conclusion already has a similar (but more focused) sentence, we decided to leave this sentence in the revised manuscript (lines 400 – 403).

Conclusions

Briefly reinforce how by addressing the DDL hypothesis in Bothrops, your work does contribute to a better understanding of the biogeography and evolution of terrestrial biodiversity in South America.

Answer: thanks for the suggestion. We tried to write a conclusion heavily focused in our results, as it already encompasses the importance of the DODL on the contribution to a better understanding of the biogeography in South America. For this, we made minor corrections in the Conclusion and decided to leave a similar conclusion in the revised manuscript (lines 406 – 411). We hope that the changes made in the discussion suggested by you support this conclusion. Thanks once again.

Figure 1 

please consider to include the letters for each biogeographical unit (in the map) in the legend. Also, you could include the phylogeny, clearly signalling which the focal forest clade is.

Answer: we did include the letters for each biogeographical unit in the legend. Thanks for this suggestion. But, as stated before, we decided to not include the phylogeny in this Figure for a better visualization. The phylogeny can be seen in S8 Fig and the distribution of all species can be seen in S3 Fig. The focal forest clade can be clearly seen in both S8 Fig and Figs 3 and 4. Thanks once again.

---

## [Decision Letter · Decision Letter 1]

16 Aug 2021

PONE-D-21-11305R1

The role of vicariance and dispersal on the temporal range dynamics of forest vipers in the Neotropical region

PLOS ONE

Dear Dr. Pontes-Nogueira,

Thank you for submitting your manuscript to PLOS ONE. After careful consideration, we feel that it has merit but does not fully meet PLOS ONE’s publication criteria as it currently stands. Therefore, we invite you to submit a revised version of the manuscript that addresses the points raised during the review process.

We look forward to receiving your revised manuscript.

Kind regards,

Tzen-Yuh Chiang

Academic Editor

PLOS ONE

Journal Requirements:

Reviewers' comments:

Reviewer's Responses to Questions

**Comments to the Author**

1. If the authors have adequately addressed your comments raised in a previous round of review and you feel that this manuscript is now acceptable for publication, you may indicate that here to bypass the “Comments to the Author” section, enter your conflict of interest statement in the “Confidential to Editor” section, and submit your "Accept" recommendation.

Reviewer #2: All comments have been addressed

2. Is the manuscript technically sound, and do the data support the conclusions?

Reviewer #2: Yes

3. Has the statistical analysis been performed appropriately and rigorously? 

Reviewer #2: Yes

4. Have the authors made all data underlying the findings in their manuscript fully available?

Reviewer #2: Yes

5. Is the manuscript presented in an intelligible fashion and written in standard English?

Reviewer #2: Yes

6. Review Comments to the Author

Reviewer #2: In this new version, Pontes-Nogueira and collaborators have successfully addressed most of my previous comments. There is a great improvement in the quality of the work in all methodological aspects, as well as in the way of transmitting the scientific reasoning, particularly in the introduction. The discussion has also been profoundly modified, now providing a good comparison of the results with the biogeography of different species in the region. My acknowledgements to the authors for such work!

I just have a few minor comments:

Line 155 - 159 – please add the number of sequences considered

Line 219-220 – please, explain better what you mean here and why you refer to Crotalus durissus

Lines 395-397 – you are not measuring or considering niche variables, so how your results could suggest anything about niche conservationism? Please, explain better.

Figs 3 and 4 – perhaps you could add a square or a lateral bar with the name of forest lanceheads after the names of the focal Bothrops to better highlight this group

7. PLOS authors have the option to publish the peer review history of their article (what does this mean?). If published, this will include your full peer review and any attached files.

Reviewer #2: No

---

## [Author Response · Author response to Decision Letter 1]

20 Aug 2021

Rebuttal Letter

 We are very thankful for the comments and revisions on our revised manuscript made by the Academic editor and reviewer #2. Below is our response to every point raised by the editor and reviewer to improve our work and to fully meet PLOS ONE’s publication criteria. We hope that we satisfyingly addressed these points and that our manuscript will now be suited for publication. 

Sincerely,

On behalf of all authors,

Matheus Pontes-Nogueira (corresponding author).

Academic Editor

Journal Requirements:

Answer: Thanks for this comment. We included 40 new citations in our manuscript in comparison to our original manuscript. However, we unfortunately forgot to mark in the ‘Revised Manuscript with Track Changes' these new references. This newer version has every new citation that we included in the revised manuscript marked accordingly. We further analysed all references to see if they meet PLOS ONE’s citation style and if they are corrected. We hope that now the reference list meets the journal’s criteria. 

Reviewer #2

In this new version, Pontes-Nogueira and collaborators have successfully addressed most of my previous comments. There is a great improvement in the quality of the work in all methodological aspects, as well as in the way of transmitting the scientific reasoning, particularly in the introduction. The discussion has also been profoundly modified, now providing a good comparison of the results with the biogeography of different species in the region. My acknowledgements to the authors for such work! I just have a few minor comments:

Answer: We are very, very thankful for this comment. We are glad to know that we encompassed all the observations that you made in the revision. We hope that once again we fully addressed all your newer comments with this revised manuscript. Thanks once again!

Line 155 - 159 – please add the number of sequences considered

Answer: thanks for the comment. We added the number of sequences considered in the text (412 sequences; line 156). Thanks.

Line 219-220 – please, explain better what you mean here and why you refer to Crotalus durissus

Answer: We elaborated further this paragraph to explain better what the maximum range size is and why we used the range size of the C. durissus in our analysis (lines 219 – 225). We thank you for this advice and comment. 

Lines 395-397 – you are not measuring or considering niche variables, so how your results could suggest anything about niche conservationism? Please, explain better.

Answer: we decided to expand this line to better explain how our results could suggest niche conservationism (lines 401-403). The lack of dispersal and the restriction to forest habitats could suggest that niche conservationism in this forest clade is important. We hope that now this statement is better explained. Thanks a lot for this comment.

Figs 3 and 4 – perhaps you could add a square or a lateral bar with the name of forest lanceheads after the names of the focal Bothrops to better highlight this group

Answer: thanks for the suggestion. We added a green square highlighting the names of the focal group. Thanks again.

---

## [Decision Letter · Decision Letter 2]

31 Aug 2021

PONE-D-21-11305R2

The role of vicariance and dispersal on the temporal range dynamics of forest vipers in the Neotropical region

PLOS ONE

Dear Dr. Pontes-Nogueira,

Thank you for submitting your manuscript to PLOS ONE. After careful consideration, we feel that it has merit but does not fully meet PLOS ONE’s publication criteria as it currently stands. Therefore, we invite you to submit a revised version of the manuscript that addresses the points raised during the review process.

We look forward to receiving your revised manuscript.

Kind regards,

Tzen-Yuh Chiang

Academic Editor

PLOS ONE

Journal Requirements:

Reviewers' comments:

Reviewer's Responses to Questions

**Comments to the Author**

1. If the authors have adequately addressed your comments raised in a previous round of review and you feel that this manuscript is now acceptable for publication, you may indicate that here to bypass the “Comments to the Author” section, enter your conflict of interest statement in the “Confidential to Editor” section, and submit your "Accept" recommendation.

Reviewer #2: All comments have been addressed

2. Is the manuscript technically sound, and do the data support the conclusions?

Reviewer #2: Yes

3. Has the statistical analysis been performed appropriately and rigorously? 

Reviewer #2: Yes

4. Have the authors made all data underlying the findings in their manuscript fully available?

Reviewer #2: Yes

5. Is the manuscript presented in an intelligible fashion and written in standard English?

Reviewer #2: Yes

6. Review Comments to the Author

Reviewer #2: In this new version, Pontes-Nogueira and collaborators have addressed all my comments. The manuscript looks fine, almost ready to be accepted for publication, except for few details in the added information about C. durissus which requires to be more specific and furthermore has a typo. Nevertheless, this is a very minor modification that can be easily solved.

Regarding this new information about C. durissus (lines 222-225):

First, the information in between () can be deleted because is repeating the information in the sentence, it does not provide any explanation. Second, I think that the authors must say something like “C. durissus is a widespread south American pit viper species showing the highest number of units within its range”. I guess, this is the reason why the authors look for other species out of the Bothrops clade to parameterise models. Finally, "specie" should be change by “species”.

7. PLOS authors have the option to publish the peer review history of their article (what does this mean?). If published, this will include your full peer review and any attached files.

Reviewer #2: No

---

## [Author Response · Author response to Decision Letter 2]

2 Sep 2021

Rebuttal Letter

 We are very thankful for the comments and revisions on our revised manuscript made by the reviewer #2. Below is our response to every point raised by the reviewer to improve our work and to fully meet PLOS ONE’s publication criteria. We hope that we satisfyingly addressed these points and that our manuscript will now be suited for publication. 

Sincerely,

On behalf of all authors,

Matheus Pontes-Nogueira (corresponding author).

Reviewer #2

In this new version, Pontes-Nogueira and collaborators have addressed all my comments. The manuscript looks fine, almost ready to be accepted for publication, except for few details in the added information about C. durissus which requires to be more specific and furthermore has a typo. Nevertheless, this is a very minor modification that can be easily solved.

Answer: thank you for these comments. We made the modifications that you suggested and hope that now the manuscript meets the journal standards. Thanks again. 

Regarding this new information about C. durissus (lines 222-225):

First, the information in between () can be deleted because is repeating the information in the sentence, it does not provide any explanation. Second, I think that the authors must say something like “C. durissus is a widespread south American pit viper species showing the highest number of units within its range”. I guess, this is the reason why the authors look for other species out of the Bothrops clade to parameterise models. Finally, "specie" should be change by “species”.

Answer: We removed the information inside (). We also wrote something similar that you suggested (lines 219 – 226). Thanks for this suggestion because we see that it needed more information about C. durissus. And finally, there was no need for the change in “specie” because we rewrote the entire sentence. Thanks a lot for the suggestions.

---

## [Decision Letter · Decision Letter 3]

6 Sep 2021

The role of vicariance and dispersal on the temporal range dynamics of forest vipers in the Neotropical region

PONE-D-21-11305R3

Dear Dr.Pontes-Nogueira,

We’re pleased to inform you that your manuscript has been judged scientifically suitable for publication and will be formally accepted for publication once it meets all outstanding technical requirements.

Kind regards,

Tzen-Yuh Chiang

Academic Editor

PLOS ONE

Additional Editor Comments (optional):

Reviewers' comments:

Reviewer's Responses to Questions

**Comments to the Author**

1. If the authors have adequately addressed your comments raised in a previous round of review and you feel that this manuscript is now acceptable for publication, you may indicate that here to bypass the “Comments to the Author” section, enter your conflict of interest statement in the “Confidential to Editor” section, and submit your "Accept" recommendation.

Reviewer #2: All comments have been addressed

2. Is the manuscript technically sound, and do the data support the conclusions?

Reviewer #2: Yes

3. Has the statistical analysis been performed appropriately and rigorously? 

Reviewer #2: Yes

4. Have the authors made all data underlying the findings in their manuscript fully available?

Reviewer #2: Yes

5. Is the manuscript presented in an intelligible fashion and written in standard English?

Reviewer #2: Yes

6. Review Comments to the Author

Reviewer #2: (No Response)

7. PLOS authors have the option to publish the peer review history of their article (what does this mean?). If published, this will include your full peer review and any attached files.

Reviewer #2: No

---

## [Editor Report · Acceptance letter]

9 Sep 2021

PONE-D-21-11305R3 

The role of vicariance and dispersal on the temporal range dynamics of forest vipers in the Neotropical region 

Dear Dr. Pontes-Nogueira:

I'm pleased to inform you that your manuscript has been deemed suitable for publication in PLOS ONE. Congratulations! Your manuscript is now with our production department. 

Kind regards, 

on behalf of

Dr. Tzen-Yuh Chiang 

Academic Editor

PLOS ONE